# 3D Chromatin Alteration by Disrupting β-Catenin/CBP Interaction Is Enriched with Insulin Signaling in Pancreatic Cancer

**DOI:** 10.3390/cancers16122202

**Published:** 2024-06-12

**Authors:** Yufan Zhou, Zhijing He, Tian Li, Lavanya Choppavarapu, Xiaohui Hu, Ruifeng Cao, Gustavo W. Leone, Michael Kahn, Victor X. Jin

**Affiliations:** 1Department of Molecular Medicine, University of Texas Health San Antonio, San Antonio, TX 78229, USA; zhouy4@uthscsa.edu (Y.Z.); xyhzj@outlook.com (Z.H.); lit5@uthscsa.edu (T.L.); 2Department of Stomatology, The Second Xiangya Hospital of Central South University, Changsha 410011, China; 3Division of Biostatistics, Medical College of Wisconsin, Milwaukee, WI 53226, USA; choppavarapu@mcw.edu; 4MCW Cancer Center, Medical College of Wisconsin, Milwaukee, WI 53226, USA; gleone@mcw.edu; 5Mellowes Center for Genomic Sciences and Precision Medicine, Medical College of Wisconsin, Milwaukee, WI 53226, USA; 6Department of Pathology, School of Basic Medical Sciences, Anhui Medical University, Hefei 230032, China; huxiaohui@ahmu.edu.cn; 7Department of Neuroscience and Cell Biology, Robert Wood Johnson Medical School, The State University of New Jersey, Piscataway, NJ 08854, USA; ruifeng.cao@rutgers.edu; 8Department of Biochemistry, Medical College of Wisconsin, Milwaukee, WI 53226, USA; 9Department of Molecular Medicine, Beckman Research Institute, City of Hope, Duarte, CA 91010, USA; mkahn@coh.org

**Keywords:** 3D chromatin architecture, β-catenin/CBP interaction, insulin signaling pathway, pancreatic cancer

## Abstract

**Simple Summary:**

Pancreatic cancer is a type of cancer with one of the highest mortality rates. The therapeutic methods for the treatment of pancreatic cancer are lacking. We found that when pancreatic cancer was treated with a small inhibitor, ICG-001, the 3D chromatin structures were altered. We further demonstrated that insulin signaling genes were significantly weakened after the treatment. We finally showed that when a gene looping structure was deleted, pancreatic cancer was impeded. Our work suggests that targeting aberrant insulin chromatin looping might be helpful for the treatment of pancreatic cancer.

**Abstract:**

The therapeutic potential of targeting the β-catenin/CBP interaction has been demonstrated in a variety of preclinical tumor models with a small molecule inhibitor, ICG-001, characterized as a β-catenin/CBP antagonist. Despite the high binding specificity of ICG-001 for the N-terminus of CBP, this β-catenin/CBP antagonist exhibits pleiotropic effects. Our recent studies found global changes in three-dimensional (3D) chromatin architecture in response to disruption of the β-catenin/CBP interaction in pancreatic cancer cells. However, an understanding of how the functional crosstalk between the antagonist and the β-catenin/CBP interaction affects changes in 3D chromatin architecture and, thereby, gene expression and downstream effects remains to be elucidated. Here, we perform Hi-C analyses on canonical and patient-derived pancreatic cancer cells before and after treatment with ICG-001. In addition to global alteration of 3D chromatin domains, we unexpectedly identify insulin signaling genes enriched in the altered chromatin domains. We further demonstrate that the chromatin loops associated with insulin signaling genes are significantly weakened after ICG-001 treatment. We finally elicit the deletion of a looping of IRS1—a key insulin signaling gene—significantly impeding pancreatic cancer cell growth, indicating that looping-mediated insulin signaling might act as an oncogenic pathway to promote pancreatic cancer progression. Our work shows that targeting aberrant insulin chromatin looping in pancreatic cancer might provide a therapeutic benefit.

## 1. Introduction

Pancreatic cancer is the tenth most common cancer in the United States, with the third highest mortality and the lowest 5-year survival rate, of less than 10% [1]. About 90% of all pancreatic cancer cases are classified as pancreatic ductal adenocarcinoma (PDAC). Since pancreatic cancer is so lethal, intensive research has been focused on identifying gene signatures to develop new therapeutic approaches. Recent genomic and epigenetic studies have revealed that epigenetic regulation is an emerging mechanism of PDAC progression [2,3,4]. Our recent studies found global changes in three-dimensional (3D) chromatin architecture, histone acetylation and methylation levels in response to the inhibition of the β-catenin/CREB-binding protein (CBP) interaction in PDAC cell lines, demonstrating the plasticity of chromosome architecture and broad epigenomic domains to epigenetic regulators [5,6]. However, the functional crosstalk between 3D chromatin architecture upon antagonizing the β-catenin/CBP interaction and critical downstream signaling pathways in pancreatic cancer is largely unknown.

Wnt/β-catenin signaling is a critical developmental signaling pathway and its dysregulation has been shown to promote development and progression in PDAC [7,8]. Wnt/β-catenin signaling inhibitors have been extensively investigated both preclinically and clinically in order to treat a variety of cancers [9,10]. A specific small molecule inhibitor, ICG-001, was initially characterized as a Wnt/β-catenin signaling antagonist due to its targeting of the CBP/β-catenin interaction and consequent indirect inhibition of CBP’s histone acetyltransferase (HAT) activity [11]. It has demonstrated in vivo activity in various preclinical cancer models and a closely related second-generation analog PRI-724 has been evaluated in a clinical setting [12,13,14]. This epigenetic modulator has been further shown to alter gene expression in various cancer cells, including pancreatic cancer cells [11,13,15], in addition to affecting the 3D genome and epigenome structure [5,6]. However, a broader exploration of the functional link between 3D chromatin architecture and Wnt/β-catenin signaling inhibitors remains to be elucidated. 

Aberrant spatial organization of chromatin has been frequently observed in various cancer cells. For example, altered chromatin architecture has been linked to breast cancer development [16] and drug resistance [17]. Cancer-specific chromatin loops have also been associated with androgen receptor activities in prostate cancer [18]. The disruption of histone H1 proteins has been found to initiate 3D chromatin remodeling that promotes lymphoma aggressiveness [19]. Inhibition of Notch signaling or the cell cycle has been shown to alter specific 3D interactions in leukemia [20]. A recent study has demonstrated that aspects of chromatin structure, including the compartment, contact domain and loop, have been globally reprogramed during pancreatic cancer metastasis [21]. However, that study did not explore 3D chromatin-mediated downstream signaling pathways or their functional relevance in pancreatic cancer. Thus, it is crucial to fully examine the roles of 3D chromatin architecture and its downstream signaling in pancreatic cancer. 

To explore the functional relevance of 3D chromatin-mediated signaling in pancreatic cancer, in this study, we conducted in situ Hi-C in canonical and patient-derived pancreatic cancer cells to examine the global 3D chromatin changes caused by the ICG-001 treatment. We then further integrated RNA-seq data to identify distinct chromatin looping genes and their associated signaling pathways. Finally, we illustrated the functional attributes of looping-mediated insulin signaling genes in pancreatic cancer progression. 

## 2. Methods

### 2.1. Cell Lines and Reagents

PANC1 (male), PATC53 (male), PATC50 (female) and HPNE (male) cells were cultured in Dulbecco’s modified Eagle’s medium (DMEM) supplemented with 10% fetal bovine serum (FBS), 2 mM L-glutamine and 1% penicillin/streptomycin until 90% confluent. Cells were kept in a cell culture incubator at 37 °C and 5% CO_2_ until the cells reached 90% confluence.

### 2.2. Cell Proliferation Assays

Cell proliferation was assessed using a Cell Counting Kit-8 (CCK-8, Dojindo Laboratories, Gaithersburg, MD, USA), following the manufacturer’s instructions. A total of 500 PATC53 cells per well were seeded in a 96-well plate. The absorbance at 450 nm was measured at Day 0, 2, 4 and 6 using a BioTek ELx800 Absorbance Microplate Reader (BioTek Instruments, Inc., Winooski, VT, USA). Before each measurement, 10 μL of CCK-8 solution was added to each well and the plate was incubated for 60 min at 37 °C.

### 2.3. Cell Migration Assays

Cell migration assays were performed with the Incucyte ZOOM Live Cell Analysis System (Sartorius Corporation, Ann Arbor, MI, USA). The protocols were executed according to the manufacturer’s user manual about 96-well scratch wound cell migration and invasion assays.

### 2.4. Cell Apoptosis Analysis

PANC1, PATC53, PATC50 and HPNE cells were sequentially treated with ICG-001 (SELLECKCHEM, Houston, TX, USA) at a concentration of 10 µM for 48 h. Caspase 3/7 activities were detected by an Apo-ONE Homogeneous Caspase-3/7 Assay kit (Promega, Madison, WI, USA) according to the manufacturer’s protocols.

### 2.5. In Situ Hi-C Profiling 

In situ Hi-C experiments were performed according to methods used in a previous study [22]. Briefly, PANC1 and PATC53 cells were sequentially treated with ICG-001 (SELLECKCHEM, Houston, TX, USA) at a concentration of 10 µM for 48 h. Cells were crosslinked with 1% formaldehyde then lysed with lysis buffer. The pelleted nuclei were digested with HindIII restriction enzyme overnight. The digested chromatin was further ligated with T4 DNA ligase after being filled with biotin-labelled dATP. Biotinylated DNA was sheared to a size of 300–500 bp and size-selected with AMPure XP beads. Biotinylated DNA was pulled down with Dynabeads MyOne Streptavidin T1 beads (Invitrogen, Waltham, MA, USA) and sheared DNA was repaired. Primers including the sequencing index were used to amplify the libraries. Both pancreatic cancer cell lines were conducted in two biological replicates. The prepared in situ Hi-C libraries were then sequenced on the Illumina HiSeq3000 platform (Illumina, Inc., San Diego, CA, USA).

### 2.6. RNA-Seq Profiling

PANC1, PATC53, PATC50 and HPNE cells were sequentially treated with ICG-001 (SELLECKCHEM, Houston, TX, USA) at a concentration of 10 µM for 48 h. Total RNA was extracted from cells with the Quick-RNA Miniprep Kit (ZYMO Research, Irvine, CA, USA) according to the manufacturer’s protocols. The RNA-seq libraries were prepared with the Ultra II Directional RNA Library Prep Kit for Illumina (NEB, Ipswich, MA, USA) according to the manufacturer’s protocols. All four pancreatic cancer or normal cell types were conducted in three biological replicates. The prepared RNA-seq libraries were then sequenced on the Illumina HiSeq3000 platform.

### 2.7. CRISPR/Cas9-Mediated Genomic Deletion

sgRNAs were cloned into lentiCRISPR v2 plasmid (Addgene, #52961) following the previously published protocol. All sgRNA sequences are listed in Appendix A. The cloned lentiviral CRISPR plasmids were verified by Sanger sequencing and co-transfected with LV-MAX packaging plasmids mix (Thermo Fisher, Waltham, MA, USA, #A43237) into 293T cells using Lipofectamine 3000 transfection reagent (Thermo Fisher, Waltham, MA, USA, #L3000015). Culture medium containing lentivirus particles was collected, filtered and used for lentivirus transfection into PATC53 cells. Empty lentiCRISPR v2 plasmid lacking expression of any sgRNAs was used as the internal control. All plasmids involved in this study are listed in Appendix A.

To delete the distal region of the IRS1 gene, two lentiCRISPR plasmids at two sites flanking the target region were co-transfected into PATC53 cells. After 24 h, the transfected PATC53 cells were selected with 2 μg/mL puromycin for 72 h and cultured to recover for 24 h. Then the cells were sorted into 96-well plates with 1 cell/well using a limited dilution method. The single cells were grown into colonies, then expanded to obtain clonal populations for further analysis. Genomic DNA from monoclonal population cells was extracted using the PureLink PCR Purification Kit (Thermo Fisher, Waltham, MA, USA, #K3100-01). Two pairs of primers were used to perform PCR amplification with Q5 High-Fidelity 2X Master Mix (NEB, #M0492S). All validation primers are listed in Appendix A. The PCR products were used to validate the deletion of the target distal region of IRS1 and the sequences were analyzed by Sanger sequencing. Monoclonal populations with a validated deleted target distal region of IRS1 were selected for the various functional assays. 

### 2.8. RT-qPCR and 3C-qPCR

Total RNA from cells was extracted using Quick-RNATM Mini Prep (Zymo Research, Irvine, CA, USA) according to the manufacturer’s instructions. Quantitative Real-time PCR was performed using a Light Cycler 480 Instrument II real-time PCR system (Roche, Porterville, CA, USA). The relative expression levels of target genes were determined by the 2^−ΔΔCt^ method, using GAPDH as an internal control. The primers are listed in Appendix A. 3C-qPCR experiments were performed as previously described [23]. The primers of 3C-qPCR are listed in Appendix A.

### 2.9. Western Blotting

Cells were lysed in RIPA lysis buffer, supplemented with 1X Halt protease and phosphatase inhibitor cocktail as well as 5 mM EDTA (Thermo Fisher Scientific, Waltham, MA, USA). Protein concentrations were quantified with the Pierce BCA protein assay kit. Next, 30 μg of protein extracts per well were loaded and separated by SDS-PAGE gels. Blotting was performed using a nitrocellulose membrane (LI-COR). After 1 h of blocking with 5% non-fat milk in TBS buffer, the membrane was incubated with primary antibody overnight at 4 °C. After washes with TBST, the membrane was incubated with IRDye 680RD Goat anti-Mouse IgG secondary antibody or 800CW Goat anti-Rabbit IgG secondary antibody (LI-COR). Signals were detected using Odyssey Imaging Systems (LI-COR) following the manufacturer’s instructions. All antibodies used in western blotting are listed in Appendix A.

### 2.10. shRNA Plasmid-Mediated Inhibition of RHBDD1 Expression

RHBDD1 and control shRNA plasmids were purchased from Santa Cruz Biotechnology, Dallas, TX, USA (sc-94654-SH, sc-108060). shRNA plasmids were transfected into PATC53 cells following the manufacturer’s protocol. At 24 h after the transfection, 2 μg/mL puromycin was used for 72 h to select the stably transfected cells. The RHBDD1 expressions of the stably transfected PACT53 cells were quantified by RT-qPCR. 

### 2.11. Hi-C Data Analysis

Raw Hi-C reads were mapped to human HG19 genome by Bowtie 2 [24]. Mapped paired reads were then filtered and normalized by HiC-Pro [25]. The compartments were identified by CscoreTool v1.1 to compute the C-score at 40 K resolution [26]. The C-score was filtered with bias > 0.2 and <5. TADs were identified by matrix2insulation.pl to produce the insulation score. The significant chromatin interaction loops were identified by HiSIF [23].

### 2.12. RNA-Seq Data Analysis

Raw RNA-seq reads were mapped to human HG19 genome by HISAT2 [27]. The expression counts were computed by htseq-count [28]. The differentially expressed genes were identified by DESeq2, with a cut-off comprising an absolute of log2 fold change ≥ 0.6 and a *p* value ≤ 0.01 [29].

### 2.13. Enrichment of Signaling Pathways

Enrichment of signaling pathways was performed with GSEA v4.1.0 on the GSEAPreranked module using the KEGG gene set database before genes were ranked by C-score differences [30].

## 3. Results

### 3.1. Pancreatic Cancer Cells Are Inhibited by Wnt Signaling Inhibitor ICG-001 

In the canonical Wnt signaling pathway, β-catenin—by binding to the members of the transcription factor T-cell factor (TCF) family—inhibits differentiation via recruitment of the histone acetyltransferase cyclic AMP response element-binding protein-binding protein (CBP) [31]. The small molecule inhibitor ICG-001 has been shown to disrupt CBP binding with β-catenin/TCF, which subsequently leads to a recruitment of the highly homologous histone acetyltransferase p300 to bind β-catenin/TCF, thereby initiating differentiation, with corresponding metabolic changes [32] (Figure 1A). We first conducted in vitro functional assays with ICG-001 on the two newly established pancreatic cancer cell lines, PATC53 and PATC50, derived from fresh patient-derived tumor xenografts [33]; the human pancreatic ductal adenocarcinoma cell line, PANC1; and the human pancreatic epithelial cell line, HPNE. We found that the growth of PANC1, PATC53, PATC50 and HPNE was significantly inhibited by ICG-001 in a dose-dependent fashion at 2, 4, 6 and 12 µM at Day 4 and Day 6 compared with the DMSO control (Figure 1B and Appendix A). Cell migration in these four cell lines was also inhibited by ICG-001 (Figure 1C and Appendix A). Importantly, cell apoptosis was significantly increased in the three pancreatic cell lines, PANC1, PATC53 and PATC50, but not in the normal HPNE cells upon ICG-001 treatment (Figure 1D and Appendix A). Since Gemcitabine, a nucleoside analogue, is a standard chemotherapeutic drug for treating pancreatic cancer patients, we also tested the efficacy of a combination of ICG-001 and Gemcitabine and found that the cell growth of pancreatic cancer cells was significantly impeded (Appendix A). Together, our results demonstrate that ICG-001 can inhibit pancreatic cancer cell growth and increase apoptosis, as has been previously observed [13]. 

### 3.2. The 3D Chromatin Architecture of Pancreatic Cancer Cells Is Altered by ICG-001

Previously, we had only investigated the global alteration of 3D chromatin domains by ICG-001 in PANC1 cells [5]. Therefore, in the present study, we were interested in investigating whether ICG-0001’s alteration of 3D chromatin was more broadly affected in pancreatic cancer. To this end, we conducted Hi-C profiling of PATC53 and PANC1 cells after 48 h of treatment with DMSO or 10 µM ICG-001 (Appendix A). We observed clear global compartment changes in PATC53 cells, as well as in PANC1 cells, after ICG-001 treatment based on C-score measurements (Figure 2A). C-scores are often used to classify the chromatin compartment type, in that a C-score larger than or equal to 0 is classified as compartment A, i.e., open and active chromatin, while a C-score smaller than 0 is classified as compartment B, i.e., closed and repressive chromatin [26]. We found that more than half of the compartments flipped after ICG-001 treatment in both pancreatic cancer cell lines (Figure 2B, Appendix A). A bird’s eye view of two genomic regions showed two types of compartment change—compartments that switched from A to B or B to A, as well as stable compartments that changed from A to A or B to B in PATC53 cells (Figure 2C). We then performed the enrichment of the KEGG signaling pathway for genes enclosed in the compartment regions, with an absolute value of a C-score difference larger than or equal to 1. Remarkably, the insulin signaling pathway was highly enriched in both PANC1 and PATC53 cells (Figure 2D), indicating that it might be a critical downstream signaling pathway regulated by 3D chromatin alteration in response to ICG-001 disruption of the β-catenin/CBP interaction in pancreatic cancer cells. 

We further examined the Topologically Associating Domains (TADs) before and after ICG-001 treatment and found that the numbers and the average sizes of TADs were essentially the same in either treated or untreated conditions (Figure 3A,B). However, many of the TADs showed shifted boundaries in both cell lines after ICG-001 treatment (Figure 3C). We then conducted RNA-seq analyses (Appendix A) and identified a total of 5667 differentially expressed genes (DEGs) in PANC1 and PATC53 cells before and after ICG-001 treatment (Figure 3D, Appendix A). Interestingly, some of these DEGs were also shown to be differentially expressed between PANC1 and HPNE, between PATC53 and HPNE, or between PATC50 and HPNE (Figure 3E, Appendix A), suggesting that some of the ICG-001-responsive genes might be directly or indirectly involved in pancreatic tumorigenesis. 

### 3.3. Insulin Signaling Pathway Is Highly Enriched in the Altered Chromatin Structure

Since insulin and insulin receptor substrate 1 (IRS1) are involved in cell proliferation, differentiation and glucose and lipid metabolism, we wanted to further examine the relationship between the insulin signaling pathway (ISP) and 3D chromatin structure in detail (Figure 4A). To this end, we examined the differences between the compartments with ISP genes and the compartments without ISP genes (non-ISP) by analyzing the C-score differences. We found that the number of compartment changes was statistically significant between ISP genes and non-ISP genes (Figure 4B). ISP genes also showed a higher shifted TAD length than non-ISP genes in both PANC1 and PATC53 (Figure 4C). We further investigated the distal–promoter looping events, where the promoter is defined as a 5 K upstream/downstream Transcription Start Site (TSS) and the distal is defined as being either 30–130 K upstream of the TSS or 5 K downstream of the TSS to Transcription Terminal Site (TTS). We identified the looping changes before and after ICG-001 treatment as the following four types: (1) ‘None to None’—no looping events either before or after ICG-001 treatment; (2) ‘Loop to Loop’—having looping events either before or after ICG-001 treatment; (3) ‘None to Loop’—looping events gained after ICG-001 treatment; and (4) ‘Loop to None’—looping events lost after ICG-001 treatment. For 137 ISP genes, we observed that there were 11 ‘None to Loop’ changes in PANC1 and 20 in PATC53, and 29 ‘Loop to None’ changes in PANC1 and 15 in PATC53 (Figure 4D, Appendix A). Together, our data illustrate that ISP is highly enriched in the altered chromatin structure. 

### 3.4. Looping Events Associated with ISP Genes Are Weakened by ICG-001

We subsequently focused on examining the ISP genes of the ‘Loop to None’ type, i.e., those with a loss of looping events after ICG-001 treatment. There are a total of thirty common ISP genes in PANC1 and PATC53 showing a loss of looping events (Figure 5A). Interestingly, six of these ISP genes were shown to be differentially expressed after ICG-001 treatment, with two up-regulated and four down-regulated (Figure 5B). The loss of looping events in these ISP genes was further confirmed by 3C-qPCR experiments on both PANC1 (Figure 5C) and PATC53 cells (Figure 5D). Remarkably, an experimentally confirmed lost loop of insulin receptor substrate (IRS1), a key ISP gene, falls within the shifted TADs in both PANC1 and PATC53 cells (Figure 5E), suggesting a functional link between ISP and the alteration of 3D chromatin domains upon the CBP/β-catenin disruption. 

### 3.5. Deletion of IRS1 Distal–Promoter Looping Impedes Pancreatic Cancer Cell Growth

To further examine the function of IRS1 distal–promoter looping, we attempted to delete the distal region of 105 K length upstream of IRS1 (Chr2: 227,692,583–227,797,349, about 29 K away from TSS of IRS1) by using CRISPR/Cas9 technology in PATC53 cells (Figure 6A). PCR products showed that the distal region on one allele was successfully deleted in two clones, Del-01 and Del-02 (Figure 6B). Sanger sequencing further confirmed such deletion of this genomic distal region (Figure 6C). The mRNA expression of IRS1 in both the Del-01 and Del-02 clones was also significantly reduced (Figure 6D). Furthermore, the protein expression of IRS1 was clearly decreased in the Del-02 clone (Figure 6E,F). Cell proliferation was also notably slowed in both the Del-01 and Del-02 clones (Figure 6G). However, the ISP proteins, including ERK1/2 (MAPK), phosphate ERK1/2, phosphate AKT and AKT, were not significantly down-regulated (Appendix A). This may be due to the fact that the level of IRS1 down-regulation was not sufficient to alter the AKT and ERK1/2 pathways. In addition, ICG-001 treatment did not have an additive inhibitory effect on Del-01 and Del-02 mutation. Since IRS1 shares the distal region with its neighboring gene RHBDD1, we wanted to rule out the idea that such aberrant pancreatic cancer cell growth was due to the alteration of RHBDD1. The mRNA levels of RHBDD1 were indeed reduced in both the Del-01 and Del-02 clones (Appendix A). The mRNA levels of RHBDD1 were also shown to be down-regulated upon the silencing of RHBBD1 (Appendix A). However, the fact that neither cell growth nor IRS1 expression was affected by shRHBDD1 (Appendix A) confirms that RHBDD1 is not involved in IRS1-regulated pancreatic cancer cell phenotypic change. Taken together, our data illustrate that looping-mediated IRS1 expression is able to regulate pancreatic cancer cell phenotypic changes. 

## 4. Discussion

Our previous investigations demonstrated that ICG-001, a specific β-catenin/CBP antagonist, was able to sensitize pancreatic cancer tumors to gemcitabine treatment [11,13] and alter chromatin architecture and broad epigenomic domains [5,6]. Another study also found that ICG-001 altered the expression of DNA replication genes and cell cycle genes, such as SKP2 and CDKN1A, in pancreatic cancer cells [12]. Our current study further systematically examined the effects of disrupting β-catenin/CBP signaling on 3D chromatin architecture in a panel of human pancreatic cancer cells. Interestingly, we identified insulin signaling as the top enriched signaling pathway in the altered chromatin structure, suggesting that insulin signaling constitutes a downstream signaling pathway regulated by 3D chromatin looping in response to ICG-001 treatment. 

Insulin signaling has been reported to promote the progression of pancreatic cancer [34,35]. Insulin receptor crosstalk with GPCRs stimulates the proliferation of pancreatic cancer [36]. The insulin-like growth factor receptor IGF1R has been found to contribute to cancer development [37], although phase III clinical trials targeting IGF1R inhibition have been unsuccessful [38,39]. One explanation could be that endocrine–exocrine signaling beyond insulin is more essential in the progression of obesity-associated pancreatic cancer [40]. Our study provides new evidence that the regulation of 3D chromatin structure and its crosstalk with insulin signaling are critical to pancreatic cancer development and progression. 

Dysregulated insulin signaling is known to be the primary cause in the development of diabetes mellitus, leading to long-term high blood sugar or hyperglycemia in both type 1 diabetes (T1D) and type 2 diabetes (T2D) [41]. Indeed, both T1D and T2D patients have a two-fold increased risk of developing pancreatic cancer [42,43]. One potential mechanism of hyperglycemia that could lead to cancer has been reported, in which high glucose levels promote the post-translational modification of O-GlcNAcylation, resulting in nucleotide imbalance, which triggers KRAS mutation in pancreatic cancer [44]. Pancreatic stellate cells normally have the ability to migrate when they are activated; however, they settle down in the pancreatic islets due to lipotoxicity in T2D [45], thereby boosting the proliferation of pancreatic cancer cells [46,47]. Our current findings may provide an opportunity to further examine the role of 3D chromatin regulation in the risk of pancreatic cancer incidence in diabetes patients. 

Although looping-mediated IRS1 has been focused on in this study, it is worth knowing that the six genes that are related to insulin signaling (Figure 5B) and show both looping and mRNA expression changes have been reported to be involved in insulin-related functions. For example, FBP1 (Fructose-Bisphosphatase 1) has a role in regulating the glucose sensing and insulin secretion of pancreatic beta cells, while PTPN1 (Protein Tyrosine Phosphatase Non-Receptor Type 1) acts as a negative regulator of insulin signaling by dephosphorylating the phosphotryosine residues of insulin receptor kinase. Therefore, in our future studies, we will examine their looping-mediated functionality in regulating pancreatic cancer progression. 

## 5. Conclusions

From the above, we can conclude that 3D chromatin alteration by disrupting the β-catenin/CBP interaction—thereby indirectly inhibiting CBP’s histone acetyltransferase (HAT) activity, with the potential recruitment of p300 HAT activity [14]—is functionally associated with insulin signaling in pancreatic cancer cells. Our data indicate that aberrant 3D chromatin-mediated insulin signaling might act as an oncogenic pathway to promote pancreatic cancer progression. Our work provides a rationale that targeting insulin chromatin looping might be a potential therapeutic strategy for treating pancreatic cancer.

## Figures and Tables

**Figure 1 cancers-16-02202-f001:**
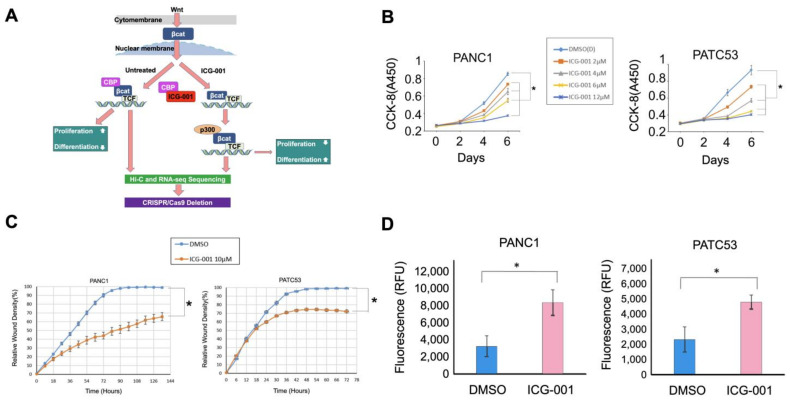
Cell growth curves and apoptosis analyses of pancreatic cancer cells during treatment with ICG-001. (**A**) Schematic view of experiments performed in this project. βcat: β-catenin; CBP: histone acetyltransferase cyclic AMP response element-binding protein-binding protein; P300: histone acetyltransferase P300; TCF: T-cell factor. (**B**) Time- and dose-dependent growth curves of PANC1 and PATC53 during ICG-001 treatment. (**C**) Cell migration assay for PANC1 and PATC53 in presence of 10 µM ICG-001. (**D**) Apoptosis analyses of pancreatic cancer cell lines PANC1 and PATC53 with ICG-001 treatment. *: samples *t*-test, *p* < 0.05.

**Figure 2 cancers-16-02202-f002:**
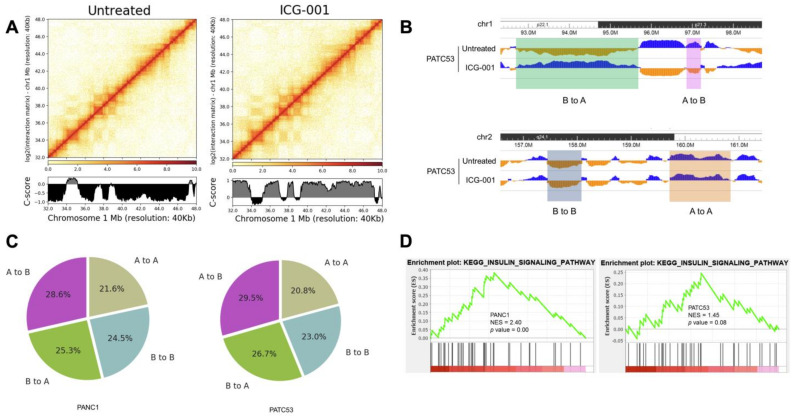
Compartment changes following ICG-001 treatment in pancreatic cancer cells. (**A**) Heatmaps and C-scores of Hi-C data in PATC53 cells. (**B**) Illustration of compartment changes. (**C**) Percentage of compartment changes both in PANC1 and PATC53. (**D**) Enrichment of insulin signaling pathway with GSEA from genes located at regions with absolute values of C-score difference greater than 1.

**Figure 3 cancers-16-02202-f003:**
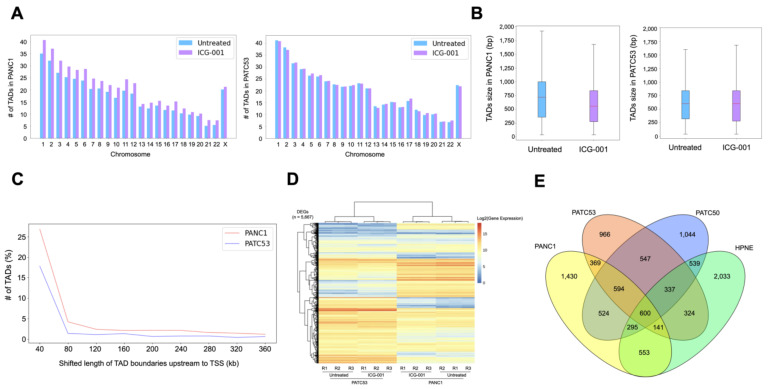
TADs and differentially expressed genes before and after ICG-001 treatment in pancreatic cancer cells. (**A**) Number of TADs in individual chromosomes in PANC1 and PATC53. (**B**) Sizes of TADs in PANC1 and PATC53. (**C**) Number of differential TADs in various shifted lengths of TAD boundaries upstream to TSS after ICG-001 treatment in PANC1 and PATC53. (**D**) Differentially expressed genes (DEGs) before and after ICG-001 treatment in PANC1 and PATC53 (n = 5667). (**E**) Venn diagram of DEGs in PANC1, PATC53, PATC50 and HPNE.

**Figure 4 cancers-16-02202-f004:**
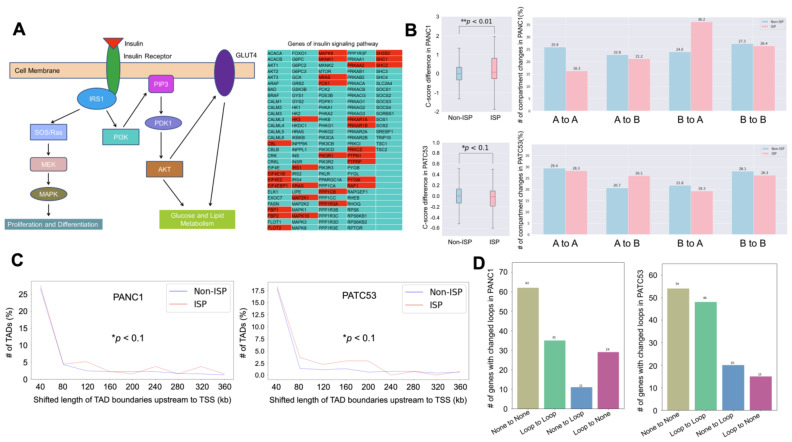
Involvement of insulin signaling pathway in changes to 3D chromatin in pancreatic cancer cells. (**A**) Overview of insulin signaling pathway (ISP). Left: major key molecules in ISP. Right: list of ISP genes—genes with missing looping events after ICG-001 treatment in PANC1/PATC53 are highlighted with red background. (**B**) Comparison of compartment changes in ISP and Non-ISP in PANC1 and PATC53. Left column: C-score difference. Right column: number of compartment changes. * and **: Wilcoxon rank-sum test. ISP: genes of insulin signaling pathway. Non-ISP: genes not in insulin signaling pathway. (**C**) Number of differential TADs, in various shifted lengths of TAD boundaries upstream to TSS. *: paired samples *t*-test. ISP: genes of insulin signaling pathway. Non-ISP: genes not in insulin signaling pathway. (**D**) Number of genes of insulin signaling pathway with changed loops in PANC1 and PATC53. None to None: no looping events with or without ICG-001 treatment. Loop to Loop: having looping events with or without ICG-001 treatment. None to Loop: looping events gained after ICG-001 treatment. Loop to None: looping events lost after ICG-001 treatment.

**Figure 5 cancers-16-02202-f005:**
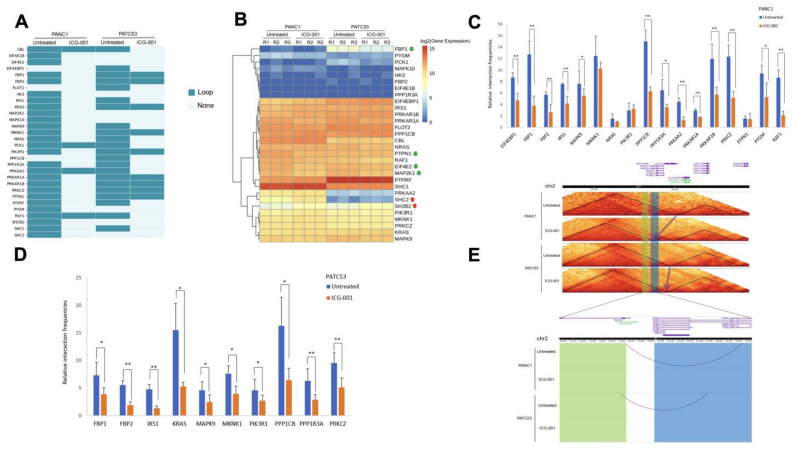
Loss of looping events after ICG-001 treatment in pancreatic cancer cells. (**A**) Loss of looping events of insulin signaling genes in PANC1 and PATC53. Loop: with looping events. None: without looping events. (**B**) Expression of genes with loss of looping events in insulin signaling pathway in PANC1 and PATC53. Red up arrow: up-regulated genes. Green down arrow: down-regulated genes. (**C**) The relative interaction frequencies of insulin signaling pathway genes in PANC1, identified by 3C-qPCR experiments. * and **: samples *t*-test, *: *p* < 0.1, **: *p* < 0.05. (**D**) The relative interaction frequencies of insulin signaling pathway genes in PATC53, identified by 3C-qPCR experiments. * and **: samples *t*-test, *: *p* < 0.1, **: *p* < 0.05. (**E**) Changes in TADs and looping events after ICG-001 treatment in PANC1 and PATC53. Up panel: TADs changes indicated by arrows. Down panel: looping events indicated by arcs. IRS1 gene body is highlighted with yellow. IRS1 30–130 K upstream to TSS is highlighted with blue.

**Figure 6 cancers-16-02202-f006:**
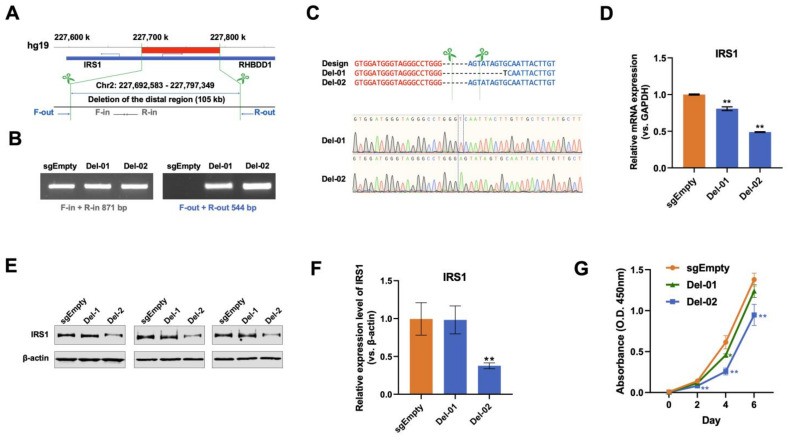
Functional characterization of IRS1 promoter–enhancer looping in PATC53 cells. (**A**) The design of CRISPR/Cas9-mediated deletion of the enhancer of IRS1. The primer pairs (F/R-out and F/R-in) were used to validate the deletion of the enhancer region. (**B**) Gel images of the PCR amplification of genomic DNA using primers inside or outside the enhancer region show the result of the enhancer deletion. Out of 73 clones, 2 clones show validated enhancer deletions. (**C**) The PCR products of the two clones (Del-01 and Del-02) were purified and Sanger sequencing was performed. Sequencing results represent the deletions induced by sgSite-01 and sgSite-02. (**D**) Quantitative PCR was performed to detect the mRNA expression levels of IRS1 in the enhancer deletion clones. (**E**) Three technical replicates of western blotting show the protein expression levels of IRS1 in the enhancer deletion clones. (**F**) The quantification of the protein expression of IRS1 in the enhancer deletion clones. (**G**) Cell growth curves show that the growth of clone Del-02 is significantly decreased compared with clone sgEmpty. *: *p* < 0.1, **: *p* < 0.05.

## Data Availability

Except the raw Hi-C data for untreated PANC1, which were downloaded from the ENCODE project using the GEO accession number GSE105566 [48], raw and processed in situ Hi-C data for DMSO or 10 µM ICG-001 two-day treated PANC1 and PATC53 cells are deposited in GEO, under accession number GSE197293. Raw and processed RNA-seq data for untreated or 10 µM ICG-001 two-day treated PANC1, PATC53, PATC50 and HPNE cells are deposited in GEO under accession number GSE198444.

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
