# Peer review of "3D Chromatin Alteration by Disrupting β-Catenin/CBP Interaction Is Enriched with Insulin Signaling in Pancreatic Cancer"

_cancers, 2024, doi:10.3390/cancers16122202_

Round 1

Reviewer 1 Report

Comments and Suggestions for Authors

Dear authors, your article is fluid and logically written with a large number of outcomes. Sometimes the scientific experiments don't provide enough evidence to back your conclusions.

In addition to just describing the genes and chromatin domains that are impacted by the treatment, I believe that you should pay attention to the function/s that these changes induce when they occur.

Here, I recommend a few fresh studies that are required to strengthen the findings and improve your conclusions.

Result 2.1. Pancreatic Cancer Cells Are Inhibited by Wnt Signaling Inhibitor ICG-001

- Assays for invasion and migration are not the same and have distinct purposes. The capacity of cells to move from an upper chamber containing a chemoattractant to a lower chamber with a micropourous membrane is known as migration. The invasion assay measures the amount of in vitro cell invasion via a layer of cells or an extracellular matrix protein.

How about your assays? Are they primarily concerned with invasion or migration? The conclusions that can be drawn varied significantly. It is necessary to distinguish them. Give specifications regarding the invasion and migration assay procedures in M&M section. 

- Please include some western blots showing at the various times the apoptosis induced upon inhibitor treatment. (Bcl-XL-PARP-Bcl-2-Caspases, etc.). Please include some immunofluorescence figures. After treatment, how is the nucleus doing? What is the state of chromatin appearance?

-Please include a few studies where the suggested combo treatment's efficacy is evident. Are the synergic? are they additive? Indicate the combination index

Result 2.5. Deletion of IRS1 Distal-Promoter Looping Impedes Pancreatic Cancer Cell Growth

IRS-1 deletion is not compleaty. There is yet a low amount of IRS detected either by wesetrn blot either by RT-PCR. Are the cells infected with lentivaral particles?

Is the population of cells ( PAT53) homogeneous? Has the Crisp Cas9 mutant been introduced into the majority of the cells?

Is IRS-1 similarly inactive in this cell polulation? Please use some IRS -1 related genes to confirm this findings. 

Comments on the Quality of English Language

Fluent in english

Author Response

Response to comments

Reviewer #1:

Comments and Suggestions for Authors

Question 1:

Dear authors, your article is fluid and logically written with a large number of outcomes. Sometimes the scientific experiments don't provide enough evidence to back your conclusions.

In addition to just describing the genes and chromatin domains that are impacted by the treatment, I believe that you should pay attention to the function/s that these changes induce when they occur.

Here, I recommend a few fresh studies that are required to strengthen the findings and improve your conclusions.

Result 2.1. Pancreatic Cancer Cells Are Inhibited by Wnt Signaling Inhibitor ICG-001

- Assays for invasion and migration are not the same and have distinct purposes. The capacity of cells to move from an upper chamber containing a chemoattractant to a lower chamber with a micropourous membrane is known as migration. The invasion assay measures the amount of in vitro cell invasion via a layer of cells or an extracellular matrix protein.

How about your assays? Are they primarily concerned with invasion or migration? The conclusions that can be drawn varied significantly. It is necessary to distinguish them. Give specifications regarding the invasion and migration assay procedures in M&M section. 

Response: Thank reviewer’s helpful suggestion. Our assay is migration assay. We updated our method on the manuscript.

Question 2:

- Please include some western blots showing at the various times the apoptosis induced upon inhibitor treatment. (Bcl-XL-PARP-Bcl-2-Caspases, etc.). Please include some immunofluorescence figures. (Include if any images are there related to the apoptosis and immunofluorescence) After treatment, how is the nucleus doing? What is the state of chromatin appearance?

Response: Our previous study has already demonstrated the western blots of the apoptosis induced by ICG-001. Please refer to Figure 5 of the following paper for the detail:

A small molecule inhibitor of beta-catenin/CREB-binding protein transcription. Emami KH, Nguyen C, Ma H, Kim DH, Jeong KW, Eguchi M, Moon RT, Teo JL, Kim HY, Moon SH, Ha JR, Kahn M. Proc Natl Acad Sci U S A. 2004 Aug 24;101(34):12682-7.

Question 3:

-Please include a few studies where the suggested combo treatment's efficacy is evident. Are the synergic? are they additive? Indicate the combination index

Response: Our previous study has already investigated the combination of ICG-001 with Gemcitabine. The combination treatment is a synergic effect because ICG-001 can dramatically sensitizes the tumor to gemcitabine, while concomitantly ameliorating gemcitabine toxicity. Please refer to the following paper for the detail:

Differentiation Therapy Targeting the β-Catenin/CBP Interaction in Pancreatic Cancer. Manegold P, Lai KKY, Wu Y, Teo JL, Lenz HJ, Genyk YS, Pandol SJ, Wu K, Lin DP, Chen Y, Nguyen C, Zhao Y, Kahn M. Cancers (Basel). 2018 Mar 29;10(4):95.

Question 4:

Result 2.5. Deletion of IRS1 Distal-Promoter Looping Impedes Pancreatic Cancer Cell Growth

 IRS-1 deletion is not compleaty. There is yet a low amount of IRS detected either by wesetrn blot either by RT-PCR. Are the cells infected with lentivaral particles?

Response: Yes, the cells were infected with lentivirus particles. In the methodology it is mentioned that sgRNAs were cloned into lentiCRISPR v2 plasmid.

Question 5:

Is the population of cells ( PAT53) homogeneous? Has the Crisp Cas9 mutant been introduced into the majority of the cells?

Response: Yes, PATC53 cells are homogeneous. The mutant has been introduced to the majority of the cells. We have used antibiotic selection and single-cell selection for mutant cloning selection. We also did PCR amplification and Sanger sequencing on the mutant cells. Please refer to Figure 6B and Figure 6C.

Question 6:

Is IRS-1 similarly inactive in this cell polulation? Please use some IRS -1 related genes to confirm this findings.

Response: We have investigated the expression levels of IRS1 related genes after IRS1 is deleted. The insulin signaling pathway proteins, including ERK1/2 (MAPK), phosphate ERK1/2, phosphate AKT and AKT, were not significantly downregulated (Figure S5A). This may be due to the fact that the level of IRS1 downregulation was not sufficient to alter the AKT and ERK1/2 pathways.

Reviewer 2 Report

Comments and Suggestions for Authors

1. The authors should report the IC50 of ICG-100 for each cell line at 24-96h.

2. The concentration of 10uM is used for migration/invasion, apoptosis and Hi-C. Given the pro-apoptotic results shown in Figure 1D, the data reported in the functional assays as well as the Hi-C data are likely to be due to cells undergoing apoptosis.

3. The efficiency of del 01 and del 02 is not convincing and their effect on the activation of AKT-ERK1/2 (western blots) is subtle. 

Comments on the Quality of English Language

1. The authors should report the IC50 of ICG-100 for each cell line at 24-96h.

2. The concentration of 10uM is used for migration/invasion, apoptosis and Hi-C. Given the pro-apoptotic results shown in Figure 1D, the data reported in the functional assays as well as the Hi-C data are likely to be due to cells undergoing apoptosis.

3. The efficiency of del 01 and del 02 is not convincing and their effect on the activation of AKT-ERK1/2 (western blots) is subtle. 

Author Response

Reviewer #2:

Comments and Suggestions for Authors

  1. The authors should report the IC50 of ICG-100 for each cell line at 24-96h

Response: IC50 for PANC1 at day 2 is 136 mM, at day 4 is 13 mM, at day 6 is 3 mM. IC50 for PATC53 at day 2 is 27 mM, at day 4 is 2 mM, at day 6 is 2 mM.

  1. The concentration of 10uM is used for migration/invasion, apoptosis and Hi-C. Given the pro-apoptotic results shown in Figure 1D, the data reported in the functional assays as well as the Hi-C data are likely to be due to cells undergoing apoptosis.

Response: Thank reviewer’s helpful comments. Yes, apoptosis could be one of factors which ICG-001 has the inhibition function of tumors. Our previous study has demonstrated ICG-001 on the role of apoptosis induction. Please refer to: A small molecule inhibitor of beta-catenin/CREB-binding protein transcription. Emami KH, Nguyen C, Ma H, Kim DH, Jeong KW, Eguchi M, Moon RT, Teo JL, Kim HY, Moon SH, Ha JR, Kahn M. Proc Natl Acad Sci U S A. 2004 Aug 24;101(34):12682-7.

  1. The efficiency of del 01 and del 02 is not convincing and their effect on the activation of AKT-ERK1/2 (western blots) is subtle. 

Response: Although the efficiency of deletions del 01 and del 02 and the observed effects on AKT-ERK1/2 activation in Western blots are subtle, our analyses indicated successful genomic modifications through PCR-based genotyping, Sanger sequencing, and qPCR validation. This subtle effect might stem from various factors, including potential compensatory mechanisms within the cellular signaling network, threshold effects inherent to signaling pathways, temporal dynamics of signaling responses, and experimental conditions.

Round 2

Reviewer 1 Report

Comments and Suggestions for Authors

Dear authors, thanks for your response.

Most of my concerns were specifically addressed. Please answer in a deeper way to the question number 6.

Question 6:

Is IRS-1 similarly inactive in this cell population? Please use some IRS -1 related genes to confirm this findings.

Response: We have investigated the expression levels of IRS1 related genes after IRS1 is deleted. The insulin signaling pathway proteins, including ERK1/2 (MAPK), phosphate ERK1/2, phosphate AKT and AKT, were not significantly downregulated (Figure S5A). This may be due to the fact that the level of IRS1 downregulation was not sufficient to alter the AKT and ERK1/2 pathways.

I would like the authours pay attention to the title of the work:3D chromatin alteration by disrupting β-catenin/CBP interaction is functionally associated with insulin signaling in pancreatic cancer.

If you are not able to demonstrate that the mechanism proposed is mediated by Insulin through IRS-1 receptor,(ERK1/2 - Akt) you cannot insert this conclusion in the title. Please add some others experiments to conclude that Insulin signaling is the mainly mechanism in these cells

Comments on the Quality of English Language

English language is fluent

Author Response

Reviewer #1:

Most of my concerns were specifically addressed. Please answer in a deeper way to the question number 6.

Question 6: Is IRS-1 similarly inactive in this cell population? Please use some IRS -1 related genes to confirm this findings.

Response: We have investigated the expression levels of IRS1 related genes after IRS1 is deleted. The insulin signaling pathway proteins, including ERK1/2 (MAPK), phosphate ERK1/2, phosphate AKT and AKT, were not significantly downregulated (Figure S5A). This may be due to the fact that the level of IRS1 downregulation was not sufficient to alter the AKT and ERK1/2 pathways.

I would like the authors pay attention to the title of the work: 3D chromatin alteration by disrupting β-catenin/CBP interaction is functionally associated with insulin signaling in pancreatic cancer. If you are not able to demonstrate that the mechanism proposed is mediated by Insulin through IRS-1 receptor, (ERK1/2 - Akt) you cannot insert this conclusion in the title. Please add some others experiments to conclude that Insulin signaling is the mainly mechanism in these cells

Response: Thanks to the reviewer’s further suggestion. We agreed our current experimental data only showed a functional link between 3D chromatin alteration and IRS1 gene, but wouldn’t strongly demonstrate its underlying mechanism that is mediated by Insulin through IRS-1 receptor. We thus toned down the claim and now changed the title to the following: “3D Chromatin Alteration by Disrupting β-Catenin/CBP Interaction Is Enriched with Insulin Signaling in Pancreatic Cancer”. In the future, we would like to comprehensively investigate the mechanism on how the alteration of 3D chromatin looping by ICG-001 is mediated by Insulin through IRS-1 receptor.

Reviewer 2 Report

Comments and Suggestions for Authors

The authors adequately addressed my previous concerns. They added experimental details and improved quality of figures.

Author Response

Thanks.